# Optical Microscopy-Guided Laser Ablation Electrospray Ionization Ion Mobility Mass Spectrometry: Ambient Single Cell Metabolomics with Increased Confidence in Molecular Identification

**DOI:** 10.3390/metabo11040200

**Published:** 2021-03-27

**Authors:** Michael J. Taylor, Sara Mattson, Andrey Liyu, Sylwia A. Stopka, Yehia M. Ibrahim, Akos Vertes, Christopher R. Anderton

**Affiliations:** 1Earth and Biological Sciences Directorate, Pacific Northwest National Laboratory, Richland, WA 99352, USA; michael.taylor@pnnl.gov (M.J.T.); Andrey.Liyu@pnnl.gov (A.L.); Yehia.Ibrahim@pnnl.gov (Y.M.I.); 2Department of Chemistry, The George Washington University, Washington, DC 20052, USA; smattson006@gwmail.gwu.edu (S.M.); sstopka@bwh.harvard.edu (S.A.S.); vertes@gwu.edu (A.V.); 3Department of Neurosurgery, Brigham and Women’s Hospital, Harvard Medical School, Boston, MA 02115, USA

**Keywords:** mass spectrometry, collisional cross section, drift tube ion mobility separation, laser ablation electrospray ionization, in situ metabolomics, ambient analysis

## Abstract

Single cell analysis is a field of increasing interest as new tools are continually being developed to understand intercellular differences within large cell populations. Laser-ablation electrospray ionization mass spectrometry (LAESI-MS) is an emerging technique for single cell metabolomics. Over the years, it has been validated that this ionization technique is advantageous for probing the molecular content of individual cells in situ. Here, we report the integration of a microscope into the optical train of the LAESI source to allow for visually informed ambient in situ single cell analysis. Additionally, we have coupled this ‘LAESI microscope’ to a drift-tube ion mobility mass spectrometer to enable separation of isobaric species and allow for the determination of ion collision cross sections in conjunction with accurate mass measurements. This combined information helps provide higher confidence for structural assignment of molecules ablated from single cells. Here, we show that this system enables the analysis of the metabolite content of *Allium cepa* epidermal cells with high confidence structural identification together with their spatial locations within a tissue.

## 1. Introduction

Molecular differences between isogenic cells occur as each cell has a unique biochemistry based on its transcript, protein, and metabolite profiles in addition to environmental perturbations [1]. Isogenic cells within tissues contain a diverse population of molecules including proteins, lipids, and metabolites, many of which may vary in their composition and concentration in response to environmental and biochemical factors [2]. Oxidative stress [3], random mutations [4], and gene fluctuations [5] can all introduce molecular profile differences between cells of the same genotype. Tissues can therefore be considered as mixtures of cells with possible subpopulations present within individual cell genotypes [6,7]. Single cell analysis is therefore vital to understand the complete biotic state of an organism given the fact that bulk analysis provides a weighted data average which obscures any cell-to-cell variance [2]. For this reason, there is increasing interest in developing tools able to characterize molecular composition on a single cell scale [2].

Mass spectrometry (MS) has become a core tool for single cell metabolomics over the last few decades [2]. This is in part due to the development of analytical probes (e.g., ion beams, lasers, and solvent junctions) capable of measuring at spatial resolutions enabling biochemical characterization at the cell level [2,8,9,10]. Additionally, instrumentation developments have expanded the detection limit for poorly ionizable and low abundant molecules providing the ability to measure sub-attomolar metabolite concentrations within individual cells [2]. For these reasons, MS is increasingly used for single cell analysis [2]. However, many MS-based methods require sample configurations, sample preparations, and analysis conditions that elicit structural and chemical changes within the cell [11,12], most prominently redistribution of labile biomolecules which include certain metabolites [13]. There are numerous examples documenting the effect of sample environment and preparation on changes in metabolite distributions within tissues across MS techniques (e.g., matrix-assisted laser desorption/ionization) [14,15]. For this reason, in situ methods, capable of native state sampling, are increasingly sought after for metabolomics.

Laser-ablation electrospray ionization mass spectrometry (LAESI-MS) is an emerging technique for in situ metabolomics that relies on mid-infrared laser irradiation (i.e., λ = 2940 nm) to probe water-rich samples. LAESI-MS requires little to no sample preparation, and therefore many biological samples can be analyzed in their native state. For this reason, LAESI-MS has been applied for metabolomics in a variety of organisms [16,17,18], and in particular, plant systems due to the availability of endogenous water within plant cells [19]. LAESI-MS operates by directing a mid-IR laser to the sample to stimulate ablation of molecules in situ, which can be performed using several sampling configurations [2]. However, fiber-based LAESI-MS (f-LAESI-MS), where the laser is guided to the sample surface via an etched optical fiber, has shown the most progress for single cell analysis [20,21,22]. f-LAESI-MS single cell studies have included microdissection for intracellular analysis [22] and identification of heterogenous cell populations [23]. However, a major limitation of ambient analysis methods, which includes LAESI-MS, is the potential for a high background interference from the surrounding environment and spectral complexity due to the lack of orthogonal separation [24]. To circumvent these challenges, high mass resolution MS has been implemented with f-LAESI-MS [23]. f-LAESI-MS combined with ultrahigh magnetic field strength Fourier transform ion cyclotron resonance (FTICR)-MS has previously been demonstrated for sensitive, high mass accuracy metabolomic applications [20]. Using this technique, isotopic fine structure patterns were resolved allowing for increased accuracy of molecular annotations within infected soybean root nodules cells [20]. However, FTICR-MS is unable to resolve isomeric species, and usually requires additional orthogonal molecular analyses (e.g., tandem MS) to annotate the detected species, which can increase analytical timescales considerably [25,26]. Accordingly, methods for rapid single cell LAESI-MS are needed.

Ion mobility separation (IMS) provides an alternative to chromatographic separation as ions can be separated according to their electrophoretic mobility in an inert carrier gas (e.g., argon, nitrogen). This method circumvents the need for lengthy chromatographic separation steps to resolve isotopologues, and is therefore is ideal for high-throughput analysis [27]. However, the most valuable aspect of IMS is the ability to measure ion neutral collision cross sections (CCS) which offers an additional orthogonal dimension that can provide higher annotation confidence in replace of tandem MS analyses [27]. Accordingly, IMS has been implemented with various ambient ionization methods in biological studies [28]. However, to this date, there have been minimal studies reporting the combination of IMS with LAESI-MS for biological studies [18,27,29,30,31,32]. Recent applications of LAESI-IMS-MS include investigating biological nitrogen fixation processes resulting from the symbiotic relationship between soil rhizobia and infected soybean root nodules [18], and coupling of a home built mid-infrared-matrix assisted laser desorption electrospray ionization (IR-MALDESI) source to a commercially available drift-tube ion mobility mass spectrometer (DTIMS) for intact protein imaging in *Quercus Alba* (oak leaf) [27].

As stated, LAESI-MS utilizes endogenous water to facilitate molecular ablation, and therefore in single cell applications, requires the IR laser probe to sample at individual cell centers with sub-cellular spot sizes [2]. For this reason, instruments combining optical microscopy with an ionization source for in situ analysis are particularly useful in this context, as optical selection of individual cells can be performed followed by MS analysis [2,33]. Additionally, dual-microscopy/MS instruments are attractive as they provide structural and chemical information from samples allowing for correlation of ion distributions with topographical features [34]. In this study, we developed a bimodal imaging system integrating a single optical train to accommodate both brightfield illumination and molecular imaging with ion mobility separation of cells in their native state within tissues (Figure 1). The use of a reflective objective in this source enables the visualization of individual cells and their targeting for LAESI-MS analysis. This LAESI-MS configuration is amenable for high-throughput single cell sampling through software automation by determining cell centroid coordinates which can be captured by the microscope software enabling sequential cell-by-cell analysis directly from tissues. Coupling this source to a DTIMS enabled ^DT^CCS values to be directly measured and cross compared against reference values from metabolite libraries allowing for higher confidence in the assignment of molecular structures from individual cells [35]. In this article, we detail the design of this “LAESI microscope,” describe the workflow for single cell analysis, demonstrate the sensitivity of our system for detection of verapamil, and finally characterize metabolites with direct ^DT^CCS measurement from 60 single *A. Cepa.* epidermal cells.

## 2. Results and Discussion

### 2.1. Optimization of the Molecular Microscope Sampling Modality

To optimize and determine the sensitivity of the DTIMS component of the LAESI microscope, droplets of verapamil were ablated at various concentrations (Appendix A) [36]. Droplets were analyzed by timing the laser firing with the ion accumulation of the DTIMS (MS triggered) and by freely firing the laser independent of the DTIMS (non-triggered). An optical image of a droplet of verapamil solution using the microscope component and an example spectrum using LAESI-DTIMS can be seen in Appendix A, respectively. A ^DT^CCS (Å^2^) of 209.3 Å^2^ ± 0.8 was determined for verapamil which is in good agreement (0.3% error) with published values (210.0 Å^2^ ± 1.1) [37]. A series of verapamil concentrations were then analyzed in both MS-triggered and non-triggered modes to determine the effect of triggering on sensitivity (Appendix A). Timing the laser to fire with the ion accumulation of the DTIMS resulted in a better sensitivity and a lower signal variability across the series of verapamil concentrations (1.0 × 10^−3^–1.0 × 10^−7^ mol/L). Conversion of abundance to signal-to-noise ratio was performed to determine the limit of detection (LOD) for both modes (Appendix A). Based on the LOD measurements, the MS-triggered mode is more sensitive than non-triggered mode (32 fmol vs. 7460 fmol). This is unsurprising as continuously pulsing the laser (non-triggered) will vary the ion yield per accumulation period, decreasing sensitivity and increasing signal variance, as noted before [25,38]. The recorded sensitivity is however lower than we have previously reported on other MS systems, where 10 fmol and 8 fmol were measured on the 21 Tesla FTICR-MS and Waters Synapt G2Si (without ion mobility active), respectively [25,39]. This decrease in sensitivity can be explained by ion losses at the DTIMS interface reducing ion transfer efficiency [40,41,42,43].

Understanding the technical noise associated with our system was imperative to accurately measure metabolic noise across a cell population [20,23]. To determine the technical noise of our system (_ηtecht_^2^), verapamil droplets (3 µL of 1.0 × 10^−3^ mol/L) deposited on parafilm coated slides were analyzed in both MS-triggered and non-triggered mode (Appendix A). Technical noise is defined as _ηtech_^2^ = _σtech_^2^/_μtech_^2^ where _σtech_^2^ is the standard deviation of the recorded abundance across the technical replicates, and _μtech_^2^ is the mean abundance. Our results demonstrated that synchronizing the laser firing with the ion accumulation of the DTIMS nearly reduced the technical noise of the system by half. Specifically, MS-triggered, and non-triggered mode resulted in a technical noise level of 0.380 and 0.603, respectively. Analysis of maltose solution droplets using MS-triggered mode was also performed which resulted in a noise level equivalent to that of verapamil droplets analyzed under the same conditions (_ηtech_^2^ = 0.361).

### 2.2. Configuration of the Optical Train for Single Cell Identification and Ablation

Laser alignment paper was used to optimize the laser focus and uniformity with respect to the sample distance to the plane of electrospray tip and MS capillary inlet by firing one laser shot across a range of Z heights (Appendix A). The optical train was then configured to perform autofocusing and large area imaging to enable at focus imaging of tissue sections larger than the diameter of the objective field of view. Incorporating a dual use reflective objective for microscopy and laser ablation allowed for simultaneous optical microscopy and LAESI-DTIMS analysis to be performed within the same optical plane without switching between objectives. Additionally, the integration of image stitching software into the system enabled control of the sample stage for sample scanning, imaging, and feature-based image recognition to recombine individual tiles into a large image and to compensate for factors such as the orientation of the sample on the stage (Figure 2). A chip with regular microwell features was used to optimize this workflow by collecting overlapping image tiles (30 µm overlap) that were intentionally misaligned in the Y axis by 100 µm (Figure 2a,b). To test the robustness of the imaging workflow, no removal of vignetting was performed on individual image tiles, as correction of vignetting increases the feature intensity at the image periphery, which improves feature based stitching [44]. This workflow was effective to compensate for image misalignment and vignetting. Recombination of separate images taken from the microwell chip based on feature recognition into a stitched 5 × 2 array is shown in Figure 2c. This workflow was then tested on single cell epidermal tissue layers of *A. Cepa*. with vignetting removal (Figure 2d) which was shown to be effective for recombining image tiles from more challenging and structurally complex *A. Cepa* tissue (Figure 2d) and was scalable to larger image arrays (e.g., tested up to a 2 × 6 array of 490 × 490 µm image tiles from *A. Cepa* tissue, as seen in Appendix A). Using the home-built calculator in our modified LabView (2019) software, pixel coordinates from the center of *A. Cepa* cells were converted to stage coordinates (Figure 2e) for LAESI-DTIMS analysis. Figure 2f shows an image of an *A. Cepa* epidermal cell following laser ablation, as evidence that ablation can be directed at individual cells.

### 2.3. LAESI-DTIMS Analysis of Single A. Cepa Epidermal Cells

LAESI-DTIMS analysis of a single *A. Cepa* epidermal cell is shown in Figure 3. The mobiligram (Figure 3c) combines a single cell spectrum (Figure 3a) with ion mobility (Figure 3b) as a visual representation of *m*/*z* versus mobility. Within the mobiligram in Figure 3c, several species can be observed in green/yellow against the electrospray background (blue). The most pronounced peaks in the corresponding overall spectrum (Figure 3a) include *m*/*z* 272.127, *m*/*z* 365.105, *m*/*z* 381.080, *m*/*z* 543.131, and *m*/*z* 705.182. We observed that these species corresponded to different monosaccharide and polysaccharide adducts based on their accurate mass measurements. These assignments are displayed in Table 1. Figure 3d displays a constrained region on the mobiligram area around the tentatively assigned trisaccharide at *m*/*z* 543.131. A study by Shrestha et al. also observed this species *m*/*z* 543.132 in *A. Cepa*, assigned to a unspecified potassiated trisaccharide [21]. C_18_ sugars present in *A. Cepa* include cellotriose, melezitose, isomaltotriose, ketose, maltotriose, and raffinose, which are indistinguishable by mass measurements alone, as they are structural isomers. To identify the specific trisaccharide in this list of isotopologues, we measured the ^DT^CCS for this peak, determining a value of 217.26 Å^2^. Comparing this value against ^DT^CCS values published in a metabolite database on a DTIMS system [35] indicates that this species is likely cellotriose. Additionally, *m*/*z* 527.158 was identified as the sodiated adduct of cellotriose, as the ^DT^CCS value (215.94 Å^2^) was in good agreement with database values (0.37% error; 216.69 ± 0.21) [35]. Observation of cellotriose a major species in *A. Cepa* is expected as cellulose is the major component of the cell wall and the predominant polysaccharide in the skins of onion [45]. These results illustrate the value of ion mobility for more confident structural assignments. Ion mobility provides an additional dimension to identify a specific trisaccharide from several possible trisaccharides. All further species putatively identified within the single *A. Cepa* epidermal cell are listed in Table 1, which include additional saccharides, secondary metabolites (e.g., indigotin, caranine, and quercetin), and flavonoids (e.g., alliin, erosone, and davidioside).

### 2.4. Mobility Separation of Isomeric Species in Single A. Cepa Epidermal Cells

DTIMS systems have the capability to resolve structural isomers, as has been well established in previous studies [46]. We observed mobility separation of two species at mass *m*/*z* 272.124 (Figure 4) with respective ^DT^CCSs of 144.22Å^2^ and 153.61 Å^2^. Shrestha et al. observed this peak in a prior study assigning it as caranine, crinine, or vittatine based on exact mass [21]. Vittatine and crinine are stereoisomers, whereas caranine is a structural isomer of vittatine and crinine, it is therefore likely that mobility separation is based on the two isomeric conformations. However, additional experiments for these metabolites are required for structural confirmation because no ^DT^CCS values are currently available in public databases [35,47]. Mobility resolution for these species was calculated R = 64.36. Our DTIMS instrument demonstrated higher mobility resolution power in comparison to our previous work on the TWIMS-based Waters Synapt G2S, which has demonstrated resolutions of R ≤ ~60 [18,48,49,50].

**Table 1 metabolites-11-00200-t001:** Putative molecular assignments from LAESI-DTIMS analysis of single *A. Cepa* epidermal cells.

Species	Structure	Observed Mass (*m*/*z*)	Expected Mass (*m*/*z*)	Deviation (ppm)	^DT^CCS (Å^2^)
Tyrosine ^1,3,5^	C_9_H_11_NO_3_ + H	182.082	182.082	0.204	(II) 146.42
Alliin ^3,4^	C_6_H_11_NO_3_S + Na	200.035	200.035	1.724	144.15
Lauric acid ^2,5^	C_12_H_24_O_2_ + H	201.184	201.184	1.827	154.01
Monosaccharide ^1,2,5^	C_6_H_12_O_6_ + K	219.026	219.026	2.077	146.42
Major (unknown)	n/a	261.144	n/a	n/a	144.15
Indigotin ^6^	C_16_H_10_N_2_O_2_ + H	263.084	263.082	7.760	145.43
Cri/(Ca/Vi) ^1,6^	C_16_H_17_NO_3_ + H	272.127	272.128	−4.900	(144.22/153.61)
Glut/cystox ^6^	C_9_H_16_N_2_O_6_S + H	281.089	281.089	−1.242	152.54
Catechin ^2,6^	C_15_H_14_O_6_ + H	291.084	291.086	−7.158	(I) 165.17
Quercetin ^6^	C_15_H_10_O_7_ + H	303.052	303.05	5.231	175.45
14-EET ^6^	C_20_H_32_O_3_ + H	321.247	321.247	1.037	181.24
Erosone ^6^	C_20_H_16_O_6_ + H	353.102	353.102	0.377	154.89
Unknown	n/a	362.097	n/a	n/a	199.11
Disaccharide ^2,5^	C_12_H_22_O_11_ + Na	365.105	365.105	1.096	(I, II) 178.48
Disaccharide ^1,2,5^	C_12_H_22_O_6_ + K	381.080	381.079	2.100	(II) 180.54
Thy-diphosphate ^5^	C_12_H_19_N_4_O_7_P_2_S	425.045	425.046	−1.178	(II) 182.21
Cya-ramnoside ^5^	C_21_H_24_O_9_ + H	434.120	434.121	−1.844	161.61
Davidioside ^6^	C_21_H_24_O_9_ + Na	443.132	443.131	1.130	169.62
Major (unknown)	n/a	524.148	n/a	n/a	201.79
Trisaccharide ^1,2,5,6^	C_18_H_32_O_16_ + Na	527.157	527.157	0.731	(I) 215.94
Trisaccharide ^1,5,6^	C_18_H_32_O_16_ + K	543.131	543.132	−1.415	(II) 217.26
Major (unknown)	n/a	558.115	n/a	n/a	213.87
Gustroside ^6^	C_27_H_34_O_14_ + Na	605.183	605.184	−2.067	194.57
AsnGluGln ^6^	C_25_H_33_N_7_O_9_ + K	614.192	614.197	−7.463	196.50
Oligosacharide ^6^	C_24_H_42_O_21_ + Na	689.210	689.211	−0.792	231.66
Major (unknown)	n/a	695.229	n/a	n/a	201.81
Oligosacharide ^1,6^	C_24_H_42_O_21_ + K	705.182	705.185	−4.930	236.69
Major (unknown)	n/a	720.192	n/a	n/a	240.71

Peak ID in database (^1^ = ref [21], ^2^ = [35], ^3^ = [51], ^4^ = [52], ^5^ = [47]), ^6^ = [Metlin]. DTCCS in database ((I) = [35], (II) = [47]).

### 2.5. Comparison of In Situ Single Cell and Bulk Analysis

To further demonstrate the benefit of in situ sampling, an *A. Cepa* extract (bulk analysis) was analyzed and compared against our in situ (single cell) analysis approach (Figure 5). Extraction was performed using the MPLEx protocol to extract metabolites from homogenized *A. Cepa* epidermal tissue as this protocol has previously been shown to be effective for the extraction of a broad range of molecular classes from plant tissues [53]. Analysis of the aqueous phase portion of the extract using our LAESI-MS microscope is shown in Figure 5. A small number of species were able to be assigned in the bulk extract (Figure 5a), which were mainly the saccharides listed in Table 1 (*m*/*z* 365.105, *m*/*z* 381.080, *m*/*z* 527.157, *m*/*z* 543.131, *m*/*z* 689.210, and *m*/*z* 705.182) whereas a greater number of lower molecular weight species were detected using an in situ single cell approach (Figure 5b). A comparison of the number of spectral features (Figure 5c), and the number of an annotated species between the bulk and single cell spectra (Figure 5d) illustrates that single cell analysis produces greater molecular coverage than bulk analysis; with saccharides being the main molecular class detected in both sampling methods. This difference in metabolite profiles can be attributed to a combination of several factors, which include dilution of specific metabolites by non-metabolite containing cells in the bulk extract [54], insolubility of metabolites detected in situ in the aqueous phase of the MPLEx extract [55], cellular heterogeneity and metabolite heterogeneity across the cell population, as well as charge competition in ESI [56].

### 2.6. Variance in Sugar Abundance within Cell Population

To demonstrate the ability of the LAESI microscope to analyze individual *A. Cepa* epidermal cells in high throughput, individual cells were optically selected, and their cell centroid coordinates were entered into the LAESI microscope. This approach was able to automatically drive the stage so that the focused laser beam was directly over each cell center and acquire a mass spectrum in situ. From our analysis, we observed that all saccharide species in Table 1 were present in this 60-cell dataset. To determine any variance in saccharide abundance (i.e., metabolic noise), background subtracted peak intensities of the mono-, di-, tri-, and tetra-saccharides were plotted against cell frequency (Figure 6). From this, variations in saccharide abundance were observed. Specifically, two different populations of cells were present based on the amount of monosaccharide (*m*/*z* 219.026) detected (Figure 6a). A bimodal distribution was observed for the monosaccharide (Figure 6a) with both gamma and gaussian distributions, whereas unimodal (gaussian) distributions were present for the other saccharides (Figure 6b–d). These results indicate that subpopulations of *A. Cepa* cells in the epidermis can be discriminated based on monosaccharide abundance (Figure 6a). The metabolic noise (_ηmet_^2^) based on saccharide abundance, ordered by the level of metabolic noise, was measured as trisaccharide (_ηmet_^2^ = 0.397, gaussian, least noise) < monosaccharide (_ηmet_^2^ = 0.410, gaussian) < disaccharide (_ηmet_^2^ = 0.445, gaussian) < tetrasaccharide (_ηmet_^2^ = 0.550, gaussian) < monosaccharide (_ηmet_^2^ = 0.566, gamma, most noise). These results indicate that the metabolic noise level (_ηmet_^2^) is greater than the technical noise level (_ηtech_^2^ = 0.380) measured for the saccharide standard, maltose (Appendix A) indicating that these results are significant.

## 3. Materials and Methods

### 3.1. Materials

Methanol, water, formic acid, verapamil, and maltose were purchased (Fisher Scientific, Hampton, NH, USA) at HPLC-grade or higher purity. Agilent Low Concentration Tuning Mix (Agilent Technologies, Santa Clara, CA, USA) was used for mass and mobility calibrations. Verapamil concentration standards for assessment of instrument sensitivity and limit of detection were generated from serial dilutions of a stock solution (1.0 × 10^−3^ mol/L). Red onion (*Allium Cepa*) was purchased from a local supermarket.

### 3.2. Instrument Design

The design of the LAESI-DTIMS instrument is illustrated in Figure 1. The XYZ stage (Zaber, Vancouver, BC, Canada) has a Peltier Plate module (Thermo Fisher Scientific, Waltham, MA, USA) that regulates the stage temperature and holds the glass mounted samples [25]. A white light source was directed through a dichroic mirror (long pass, cut-on wavelength, 605 nm, Thorlabs Inc, Newton, NJ, USA) and then through a reflective objective (×15, Thorlabs Inc, Newton, NJ, USA), and a CCD (Thorlabs Inc., Newton, NJ, USA) mounted at the top of the optical stack recorded images of the sample surface. The microscope control software (µManager [58]) allowed for the specification of the number of image tiles (stage positions) to record. µManager moved the stage to each tile position, performed autofocusing by moving the stage Z axis by specifying large (10 µm × 20 steps) and fine adjustment (0.1 µm × 20 steps) over a range of Z heights. By recording images at each Z location, it selected the image with the best focus based upon its sharpness metrics [58]. The stage then moved to the next position and repeated the process. Images were compiled and image arrays were extracted into a separate processing program (Fiji) [59]. The rolling ball BASIC plugin [60] removed vignetting in individual tiles, then stitching was performed by specifying a maximum percentage overlap (18% overlap) to shift images laterally and horizontally. Image tiles were combined into an array with X/Y number of pixels. Pixel coordinates were converted to stage coordinates by inputting pixel positions in the stitched image array into a home-built calculator (developed in LabView) [61] and by specifying size dimensions (µm) of individual image tiles. Stage positions for individual cell centers in the large area stitch were then manually selected, and a list of cell center positions was exported as a .csv file format. The cell center list was uploaded into our home-built LabView-based LAESI software [61], which moved the stage to each position sequentially for LAESI-DTIMS analysis. The mid IR laser source (λ = 2940 nm; OPOTEK, Carlsbad, CA, USA), was directed through a reflective objective, through an optical train that contained the dichroic mirror (long pass, cut-on wavelength, 605 nm, Thorlabs Inc, Newton, NJ, USA) that the white light source passed through, and this permitted in-line optical microscopy. The DTIMS contained a 79 cm length linear drift cell and coupled to a quadrupole time-of-flight (QTOF) mass spectrometer (6538 QTOF, Agilent Technologies, Santa Clara, CA, USA). This systems has been described previously in detail [62]. For all experiments the following parameters were used for the DTIMS: inlet capillary (370 V), high pressure funnel entrance (360 V), high pressure funnel exit (170 V), ion funnel trap entrance (200 V), ion funnel trap exit (30 V), ion funnel trap conductance limiting orifice voltage (10 V), drift tube entrance (1900 V), drift tube exit (354 V), rear ion funnel entrance (340 V), rear ion funnel exit (160 V), short quadrupole bias (116 V), conductance limiting orifice (114 V), segmented quadrupole entrance (110 V), segmented quadrupole exit (30 V), and conductance limiting orifice (10 V). Nitrogen was used as a buffer gas in the drift tube. The pressure in high pressure funnel (3.77 Torr), Ion funnel trap (4.56 Torr), and short quadrupole (454 mTorr). The drift cell temperature was 26 °C. Multiplexing was used. MS-dependent laser triggering is described in detail elsewhere [25].

### 3.3. Sensitivity Measurements

Verapamil hydrochloride standard solutions were prepared in nano-pure deionized water (18.2 Ω). The series of concentration standards used to assess sensitivity were prepared by serial dilution of a (1.0 × 10^−3^ mol/l stock solution) to generate solution concentrations of 1.0 × 10^−3^, 1.0 × 10^−4^, 1.0 × 10^−5^, 1.0 × 10^−6^, and 1.0 × 10^−7^ mol/L. 3 µL droplets from each solution were pipetted onto a parafilm covered microscope slide. The droplets were ablated using the mid IR laser source either in sync with the ion trap of the MS opening (MS-triggered mode) or by freely firing the laser (non-triggered mode). On average, 60 laser shots were required to ablate the entire droplet. 10 technical replicates were taken for each concentration. For MS-triggered mode, the laser is externally triggered via the custom built LabView Software, and a Q-switch delay of 665 µs, pulse rate of 20 kHz, and an accumulation time 50 ms were used. For non-triggered mode, the laser was internally triggered, and a pulse rate of 20 kHz and a Q-switch delay of 165 µs was used. Note, there is a ~500 µs delay in externally triggering the flash lamp, which is why there is a difference in the Q-switch delay between MS-triggered and non-triggered modes. The laser power exiting the laser was measured as 6.166 mJ (RSD = 4.59%) using a C-series laser power meter (Thorlabs Inc, Newton, NJ) using this Q-switch delay. However, since the laser is attenuated as it passes through the dichroic mirror, the laser power was approximated to be 1.065 mJ prior to passing through the dual objective lens. This laser power was maintained for all experiments.

### 3.4. Optimizing LAESI-DTIMS Analysis

*A. Cepa* epidermal tissue mounted on a glass microscope slide was placed on the Peltier stage at 16 °C. The sample stage was leveled prior to this using a bubble level ruler (Johnson Level & Tool Mfg. Co., Inc., Mequon, WI, USA). An electrospray (+1635 V) of (1:1 MeOH/water with 0.1% formic acid was applied orthogonal to the stage surface at a constant flow rate (0.8 µL/min), and the instrument-inlet capillary voltage was held at +370.29 V. The electrospray tip was placed 9 mm from the MS inlet. The sample was then moved in Z direction to 6.5 mm below the MS inlet capillary and the microscope objective was moved in Z to bring the *A. Cepa* cells into focus. Ion transmission efficiency was maximized by freely firing the laser in non-triggered mode at 10 kHz (Q-switch delay of 165 µs), while moving the stage in the Z axis by 0.1 mm increments to maximize the total ion count (TIC_max_). The sample height was recorded at the TIC_max_, then the objective was repositioned in Z to bring the image into focus. The sample stage was moved to adjacent locations, and the TIC was measured in triggered mode (20 kHz, 50 ms accumulation time, 2 laser pulses) moving the stage in Z by 0.1 mm increments between adjacent positions to optimize ion transmission which was recorded as a sample height of 6.13 mm from the MS inlet capillary to the sample surface.

ZAP-IT laser alignment paper (Concord, NH, USA) was then placed on the sample stage, and one laser shot was fired over the range of sample Z heights (6.01–6.18 mm, distance from MS inlet to sample surface) at adjacent areas on the ZAP-IT paper. Images of these areas were recorded with the CCD, and the ratio of width to length for the impact craters was compared (i.e., ratio = 1 is ideal for a TEM_00_ beam), which was determined to be a stage Z height of 6.06 mm (distance from MS inlet to sample surface). This information is shown in Appendix A. Onion cells on microscope slides were then placed back on the stage, and the laser was fired at a Z stage height of 6.06 mm (best laser focus) and the total ion count observed, and impact crater was measured to ensure ablation of singular cells.

### 3.5. A. Cepa Cell Analysis

Spectra with mobility analysis were collected from 60 individual red onion (*Allium Cepa*) epidermal cells in positive ion mode. Briefly, 3 cm^2^ squares from the 2nd layer of an *A. Cepa* bulb were placed on a microscope slide, then the slide was placed on the Peltier-cooled stage (16 °C). An image of the *A. Cepa* tissue was recorded with the optical microscope component, then individual cell centers were manually selected, and exported into the cell center list for LAESI-MS analysis. Electrospray (+1635 V) of 1:1 MeOH/water with 0.1% formic acid was applied orthogonal to the stage surface at a constant flow rate (0.8 µL/min), and the instrument-inlet capillary voltage was held at +370 V. The number of laser pulses (*n* = 2), number of sample positions, the laser Q-switch delay, and pulse frequency (20 kHz) were all set in the custom LabView software [25]. Q-switch delay of 665 µs, pulse rate of 20 kHz, and an accumulation time 50 ms was used for all single cell experiments. The stage height (z) was maintained at 6.13 mm (distance from the MS inlet capillary to the sample surface) throughout the separate measurements. Three independent tissue sections, 30 individual cells in each of the section were measured representing a sample set of *n* = 90. This sample set was filtered down to 60 cells following LAESI-DTIMS analysis by inspecting the impact craters using our optical train. Filtering was based on the precision of the ablation spot to the cell center, as a number of the cells analyzed presented with impact craters near the cellular grain boundary which would introduce potential error into the single cell dataset.

### 3.6. MPLEx Preparation and Analysis

MPLEx extraction was performed as described in the literature [53]. Briefly, homogenized *A. Cepa* tissue (100 mg) was resuspended in water, and 5 volumes (0.2 mL) of cold (20 °C) chloroform-methanol (2:1/*v*:*v*) solution was added. Samples were incubated for 5 min on ice, vortex mixed for 1 min, and centrifuged at 12,000 rpm for 10 min at 4 °C. The upper aqueous phase containing the hydrophilic metabolites were collected in glass autosampler vials. Analysis was performed by dispensing 3 µL droplets of the metabolite extract onto parafilm coated microscope slides. On average 60 laser shots were required to ablate the entire droplet, and 10 technical replicates were analyzed of these samples.

### 3.7. Data Analysis & Software

All spectra were processed using the Pacific Northwest National Laboratory (PNNL) unified ion mobility format (UIMF) viewer [63]. Calibration of mobility to ^DT^CCS was performed with Agilent tune mix (*m*/*z* 118.086, *m*/*z* 332.048, *m*/*z* 622.029, *m*/*z* 922.010, *m*/*z* 1221.991, *m*/*z* 1521.971). Calibrated spectra were exported into mMass (open source software) for data processing [64]. Technical and metabolic noise with distribution fitting was performed in OriginPro (version 2021, OriginLab Corporation, Northampton, MA, USA).

## 4. Conclusions

Here, the development of a LAESI-IMS system integrated with an optical microscope for ambient sampling of in situ single cells is presented. This source configuration is effective for analyzing *A. Cepa* cells in a high throughput approach. Specifically, the addition of the cell recognition software, wherein spatial coordinates are assigned to individual cells, can be utilized for rapid single cell analysis. Moreover, dual channel imaging can be achieved with the system by integrating additional optics and cameras (i.e., charge coupled devices) to permit multiwavelength microscopy (e.g., brightfield plus fluorescence). Additionally, the software integrated into this LAESI microscope is all open source, therefore each individual software package can be more readily combined into a dual LAESI/microscope due to the availability of the source code. The final benefit of our LAESI microscope is that the source component can be adapted to various mass spectrometers, meaning that higher mobility resolution methods can be explored for improved separation of isotopologues within single cells [57]. The cells we analyzed here were relatively large in comparison to smaller cell sizes (e.g., mammalian). As such, in further studies we will explore higher resolution optics for improving spatial resolutions, as well as look to enhance the sensitivity of our system by investigating methods to increase ion transmission efficiency, all in effort to make single cell analysis of smaller cells possible.

## Figures and Tables

**Figure 1 metabolites-11-00200-f001:**
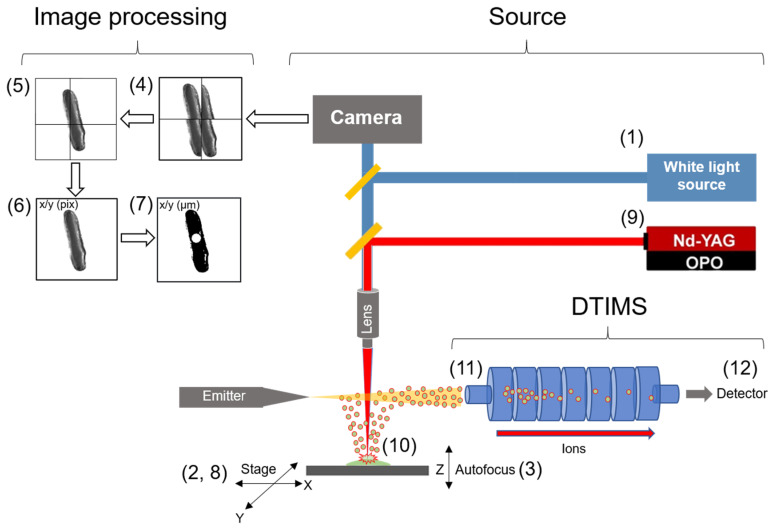
Schematic of LAESI-DTIMS based microscope system and its use. (1) A white light source is directed to the sample surface. (2) The Peltier-cooled stage moves to a sample position. (3) The microscope software performs autofocusing at the sample position by fine adjustment of the stage in the Z axis and records an image, then the stage moves in X/Y to an adjacent area to collect an image array with a specified number of tiles. (4) Image tiles are aligned. (5) Image tiles are overlapped and stitched together. (6) image coordinates (pixels) are converted to stage coordinates (µm) using a home built LabView software. (7) Cell center coordinates (X/Y µm) are extracted, and a list of cell centers coordinates are entered into the LAESI-MS microscope software. (8) The stage moves sequentially to each cell centroid coordinate. (9) When the stage arrives at a cell centroid the MS sends a trigger to the laser to fire upon the ion trap opening. (10) The ion trap opens, the targeted cell is ablated, and endogenous molecules are intercepted by the electrospray and directed into MS inlet. (11) Ion packets are accumulated in the quadrupole trap and injected into DTIMS. (12) Ions are separated in the gas phase according to electrophoretic mobility and mass over charge of each ion is recorded by the TOF-MS.

**Figure 2 metabolites-11-00200-f002:**
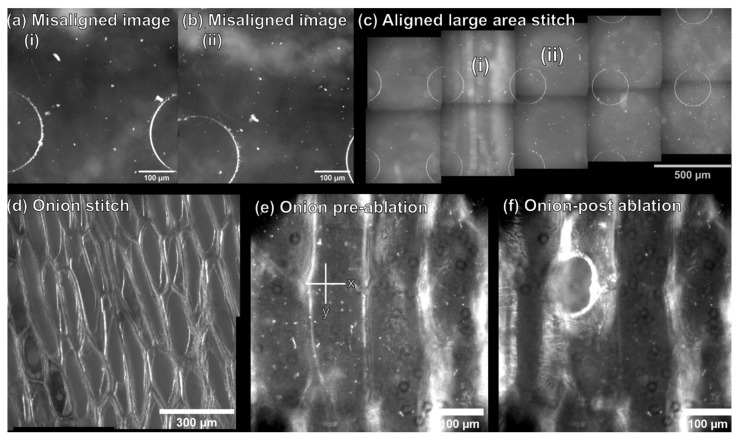
Optimizing the optical microscopy component for feature-based image stitching and single cell identification. (**a**,**b**) Separate image tiles from a microwell chip with vertical misalignment (100 µm), where (**c**) the misaligned tiles (i) and (ii) are recombined into an aligned stitched mosaic image based on feature recognition. (**d**) Image tiles from *A. Cepa* tissue are aligned and stitched together into a 2 × 2 stitched mosaic image. (**e**) An individual *A. Cepa*. epidermal cell prior to laser ablation, cell coordinates are extracted as stage coordinates (X, Y; µm, µm), and (**f**) the same single cell after LAESI-DTIMS sampling.

**Figure 3 metabolites-11-00200-f003:**
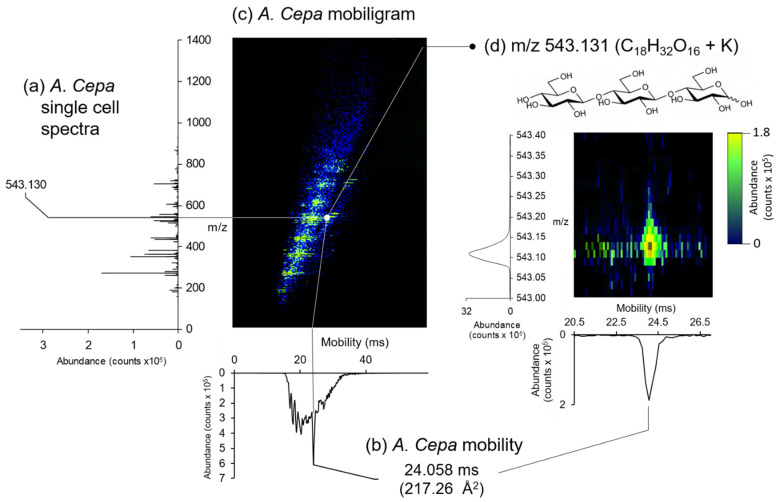
Example LAESI-DTIMS analysis of a single *A. Cepa* cell. (**a**) Spectrum with the saccharide peak labeled, (**b**) the corresponding mobility graph, and the (**c**) mobiligram showing detected species (green/yellow). (**d**) Spectrum and mobility graph of *m*/*z* 543.131 (C_18_H_32_O_16_ + K, Trisaccharide) with corresponding mobiligram.

**Figure 4 metabolites-11-00200-f004:**
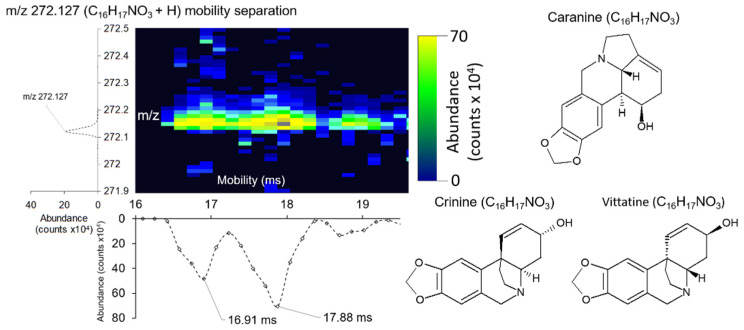
Example of how DTIMS is able to separate structural isomers in a single *A. Cepa* cell. Mobility separation of peak at *m*/*z* 272.127 (C_18_H_17_NO_3_ + H) into two species tentatively assigned as caranine (^DT^CCS = 144.22 Å^2^) and crinine/vittatine (^DT^CCS = 153.61 Å^2^).

**Figure 5 metabolites-11-00200-f005:**
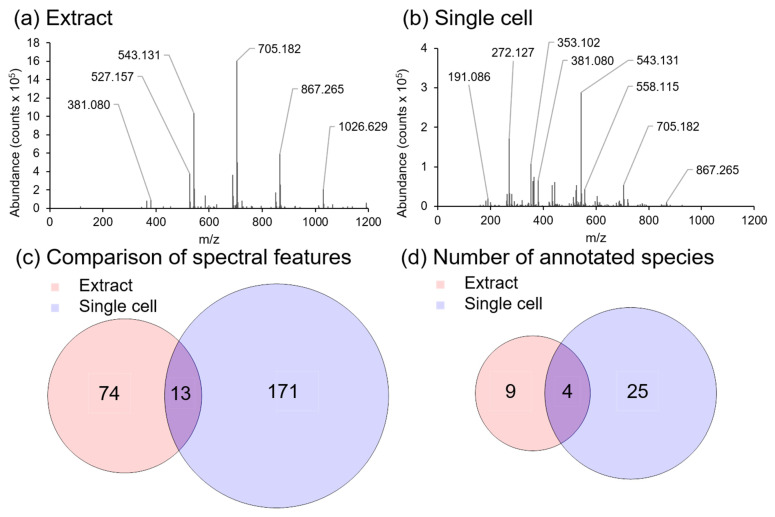
Spectral comparison of an aqueous extract from homogenized *A. Cepa*, and a single *A. Cepa* epidermal cell with LAESI-MS. (**a**) Positive ion spectrum acquired by ablation of a 3 µl droplet of an *A. Cepa* tissue extract on a parafilm coated microscope slide with LAESI-MS, and (**b**) positive ion spectrum of an *A. Cepa* single epidermal cell. (**c**) Venn diagram showing the number of unique and shared spectral features between the aqueous extract and the single cell spectra. (**d**) Venn diagram showing the number of unique and shared annotated species between the aqueous extract and the single cell spectra.

**Figure 6 metabolites-11-00200-f006:**
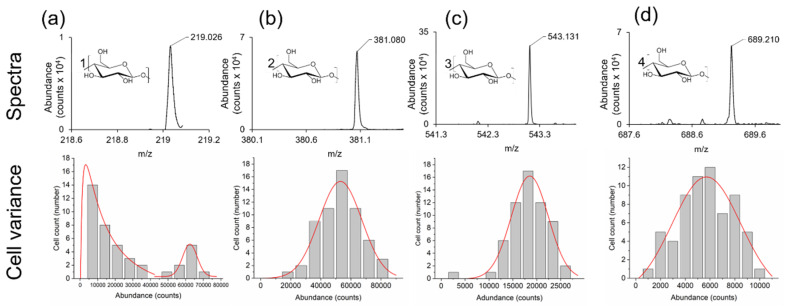
Exploring the variance in the abundance of saccharides within single cells of *A. Cepa*. i = 60 single cells. Monosaccharide (**a**), disaccharide (**b**), trisaccharide (**c**), tetrasaccharide (**d**). Spectra and structures for each of the saccharides (top row), and abundance (counts) for the saccharides within the 60 cells set (bottom row). Distribution fits are also shown (red lines).

## Data Availability

The data presented in this study are available in the Appendix A.

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
