# Peer review of "Optical Microscopy-Guided Laser Ablation Electrospray Ionization Ion Mobility Mass Spectrometry: Ambient Single Cell Metabolomics with Increased Confidence in Molecular Identification"

_metabolites, 2021, doi:10.3390/metabo11040200_

Round 1
Reviewer 1 Report
The article "Optical Microscopy-guided Laser Ablation Electrospray Ionization Ion Mobility Mass Spectrometry: Ambient Single Cell Metabolomics with Increased Confidence in Molecular Identifications" summarizes the work of Taylor et al. on performing single cell measurements of Allium cepa epidermal cells for their metabolic profiles. Ultimately, I think this is a great piece of literature that will be of interest to the field of single cell mass spectrometry. The manuscript is well written and the figures are great! The LAESI microscope is also a clever way of automating single cell analysis. I believe the work should be published and will be of great interest to the community. With that, I think there are are a couple areas that could be improved as detailed below: Major Comments: 1. The first paragraph of the introduction discusses the importance of single cell measurements but I am left convinced (although I know single cell analysis is important). I think additional references covering the functional and/or drug response ramifications of single cell subpopulations would help (10.1016/j.stem.2015.04.004, 10.1126/science.aau5324, 10.1126/science.1198704). I only list three examples here, but there are many that exist to support the argument. This or other references can be used. 2. The introduction brushes over single cell mass spectrometry measurements that have been made in the past that are fundamental to the field's progression. While space can be tight, 15/59 (25%!) of the references are related to the Vertes lab. While the Vartes lab has done amazing work in this field, I think a better distribution could be attained. Here are several references the authors should consider including to be more representative (10.1039/C0LC00211A, 10.1073/pnas.1423682112, 10.1021/acs.analchem.7b00880, 10.1021/ac102702m, 10.1021/ac1015326, 10.1002/ange.201812892, 10.1021/acs.analchem.9b01689, 10.1038/nmeth.4071, 10.1038/s41592-019-0536-2, 10.1021/ac9701748, 10.1021/acs.analchem.7b01436). One of the most notable exemptions is microscopy-guided mass spectrometry for single cell analysis (10.1021/jasms.8b05643). 3. One of the arguments the authors make is that they can perform single cell measurements with minimal sample preparations. This is often beneficial, but some of the sample preparations prevent metabolite degradation and metabolite diffusion (e.g. MALDI matrix is an antioxidant, SIMS sample drying and low source pressure prevent diffusion and degradation). This is particularly problematic for small molecule analysis and even more when performing single cell measurements. Did the authors monitor sample degradation through their experiment? Or did they address this in a different way? It is possible the analysis is fast enough it may not be an issue, but I couldn't find speed or throughput measurements. 4.First sentence of section 2.2 is not representative of the single cell MS field (see above references), so I would just remove it or explain how it is important for your specific experiment. Often the probe is larger than the cell for complete cellular analysis. 5. There is a harsh mass cut off below an m/z value of 200. Why is that? There are some important small molecules below that that may be of interest. (to clarify, I don't think the authors need to fix this mass cut off, per say, but addressing it would be helpful). 6. The is a mass error spread of 36.58 ppm (-17.18 to 19.4 ppm). This seems high for the system they are using, particularly since there are mass assignments with substantially smaller mass errors (Author Response
The article "Optical Microscopy-guided Laser Ablation Electrospray Ionization Ion Mobility Mass Spectrometry: Ambient Single Cell Metabolomics with Increased Confidence in Molecular Identifications" summarizes the work of Taylor et al. on performing single cell measurements of Allium cepa epidermal cells for their metabolic profiles. Ultimately, I think this is a great piece of literature that will be of interest to the field of single cell mass spectrometry. The manuscript is well written, and the figures are great! The LAESI microscope is also a clever way of automating single cell analysis. I believe the work should be published and will be of great interest to the community. With that, I think there are a couple areas that could be improved as detailed below:
We thank the Reviewer for their positive feedback!
Major Comments:
- The first paragraph of the introduction discusses the importance of single cell measurements, but I am left convinced (although I know single cell analysis is important). I think additional references covering the functional and/or drug response ramifications of single cell subpopulations would help (10.1016/j.stem.2015.04.004, 10.1126/science.aau5324, 10.1126/science.1198704). I only list three examples here, but there are many that exist to support the argument. This or other references can be used.
We agree and have added references (10.1016/j.stem.2015.04.004) and 10.1126/science.1198704 to line 35 as more examples.
- The introduction brushes over single cell mass spectrometry measurements that have been made in the past that are fundamental to the field's progression. While space can be tight, 15/59 (25%!) of the references are related to the Vertes lab. While the Vertes lab has done amazing work in this field, I think a better distribution could be attained. Here are several references the authors should consider including to be more representative (10.1039/C0LC00211A, 10.1073/pnas.1423682112, 10.1021/acs.analchem.7b00880, 10.1021/ac102702m, 10.1021/ac1015326, 10.1002/ange.201812892, 10.1021/acs.analchem.9b01689, 10.1038/nmeth.4071, 10.1038/s41592-019-0536-2, 10.1021/ac9701748, 10.1021/acs.analchem.7b01436). One of the most notable exemptions is microscopy-guided mass spectrometry for single cell analysis (10.1021/jasms.8b05643).
We thank the Reviewer for their insight. We have added references 10.1021/ac1015326, 10.1002/ange.201812892, 10.1021/acs.analchem.9b01689 (line 41), and 10.1007/s13361-017-1704-1 (line 83) as a nod to those that have made significant contributions in the field.
- One of the arguments the authors make is that they can perform single cell measurements with minimal sample preparations. This is often beneficial, but some of the sample preparations prevent metabolite degradation and metabolite diffusion (e.g., MALDI matrix is an antioxidant, SIMS sample drying, and low source pressure prevent diffusion and degradation). This is particularly problematic for small molecule analysis and even more when performing single cell measurements. Did the authors monitor sample degradation through their experiment? Or did they address this in a different way? It is possible the analysis is fast enough it may not be an issue, but I couldn't find speed or throughput measurements.
We agree, the Reviewer brings up a valid point related to sample change in response to analysis conditions. In the materials and methods section, we state that the sample stage temperature is held at 16 °C throughout the measurement to limit any diffusion or sample degradation. We also agree with the statement that throughput speed is an important factor, however in this paper we manually selected individual cells, meaning that the single cell analysis speed is heavily dependent on the speed of cell selection from the user. As such, adding parameters related to speed would be erroneous. In the methods section we do state the number of cells in the sample set, and the fill time (50 ms) per cell.
4.First sentence of section 2.2 is not representative of the single cell MS field (see above references), so I would just remove it or explain how it is important for your specific experiment. Often the probe is larger than the cell for complete cellular analysis.
We agree and have removed the first sentence (line 129-130).
- There is a harsh mass cut off below an m/z value of 200. Why is that? There are some important small molecules below that that may be of interest. (to clarify, I don't think the authors need to fix this mass cut off, per say, but addressing it would be helpful).
The mass cutoff is related to the RF parameters in the ion funnels at the quadrupole. Unfortunately, we encountered issues with tuning the funnel/quad for the low mass range. We hope to address lower mass range analysis in following publications.
- The is a mass error spread of 36.58 ppm (-17.18 to 19.4 ppm). This seems high for the system they are using, particularly since there are mass assignments with substantially smaller mass error
We appreciate the with the Reviewer’s eye for this detail. We found a minor calibration error with the standard (verapamil) used to calibrate our datasets. The error spread has been reduced from 36.58 ppm to 15.22 ppm. This is within the expected range for our DTIMS-TOF-MS system. Again, we thank the reviewer for highlighting this. We corrected observed mass, deviation, and measured CCS in Table 1 accordingly.
Reviewer 2 Report
Taylor et al. presents a single-cell application of optical-microscopy guided laser ablation electrospray ionization (LAESI) coupled with drift-tube ion mobility spectrometry mass spectrometry (IMS-MS). The authors performed analysis of 60 single Allium cepa (onion) epidermal cells. There is a lot to like in this manuscript. The authors carefully described the workflows, namely the triggered and non-triggered (synchronized and asynchronized?) approaches and optical modifications/image stitching in order to collect single-cell data. Analytical figures of merits were properly reported. In terms of experimental findings, the authors focused on two examples: (a) the trisaccharide at m/z 543 and (b) the caranine, crinine, or vittatine isomers; both are interesting. The rationale/explanation is good. A list of putative metabolites was also reported. I believe this manuscript can be published after a minor revision. Below are my comments:
1- Plant cell is much larger in size than mammalian cells. The title of the paper should be specific. If not, the authors should discuss what could be the smallest single cell that can be analyzed by this technique.
2- About the triggered vs. non-triggered approaches, could the authors describe how they time laser firing and ion accumulation? Is there a hardware modification that allows for the ion pulse from the entrance funnel to the drift cell to trigger a laser pulse? Or the authors wait until the ion signal depleted before firing the laser again? Did the authors use multiplex? The authors did report the main parameters in the method section, but I hope that they can elaborate on these parameters a little more.
A minor comment: the authors may consider reporting the instrument parameters as suggested by Gabelica et al. (10.1002/mas.21585).
3- Can the authors perform a more in-depth analysis of the data? In the comparison between single-cell and bulk cell analyses, the authors stated that a lot more molecules were detected in single cell experiments. Is there a way to represent them qualitatively? I.e., the average number of molecules per single-cell mass spectrum vs. the number of molecules detected in bulk analysis? A Venn diagram would be nice but the authors can choose to do it differently. Same comment can be made on last section on variations in the amount of saccharides. Can the authors make a 2D plot of intensity vs. cell size?
4- Could some of the putative metabolites, or the isomers in Figure 4 be confirmed by LC-MS/MS of bulk homogenates/extracts?
5- There is a problem with the texts in the first paragraph of section 2.3. A few sentences/phrases keep repeating.
Author Response
Taylor et al. presents a single-cell application of optical-microscopy guided laser ablation electrospray ionization (LAESI) coupled with drift-tube ion mobility spectrometry mass spectrometry (IMS-MS). The authors performed analysis of 60 single Allium cepa (onion) epidermal cells. There is a lot to like in this manuscript. The authors carefully described the workflows, namely the triggered and non-triggered (synchronized and asynchronized?) approaches and optical modifications/image stitching in order to collect single-cell data. Analytical figures of merits were properly reported. In terms of experimental findings, the authors focused on two examples: (a) the trisaccharide at m/z 543 and (b) the caranine, crinine, or vittatine isomers; both are interesting. The rationale/explanation is good. A list of putative metabolites was also reported. I believe this manuscript can be published after a minor revision. Below are my comments:
We thank the Reviewer for their encouraging feedback and suggestions.
1- Plant cell is much larger in size than mammalian cells. The title of the paper should be specific. If not, the authors should discuss what could be the smallest single cell that can be analyzed by this technique.
We agree with the Reviewer’s statement related to differences in cell sizes. We have added a statement in the conclusion section (line 295-300) reflecting the limitations of our LAESI system for analysis of small cells.
2- About the triggered vs. non-triggered approaches, could the authors describe how they time laser firing and ion accumulation? Is there a hardware modification that allows for the ion pulse from the entrance funnel to the drift cell to trigger a laser pulse? Or the authors wait until the ion signal depleted before firing the laser again? Did the authors use multiplex? The authors did report the main parameters in the method section, but I hope that they can elaborate on these parameters a little more.
We thank the Reviewer for the comment, as it is good to make sure there is clarity for this vital point of the study. This system has been described in great detail in previous publications as we state in the manuscript (line 331, reference 62 Anal. Chem. 2019, 91, 5028–5035. 10.1021/acs.analchem.8b05084.). Briefly, the mass spectrometer sends a TTL pulse when the ion trap is ready to accumulate ions. This trigger initiates the flashlamp of the laser to fire followed by a second trigger with a delay time to open the Q-switch a deliver a laser pulse to the sample surface. Ions are accumulated for 50 ms, following which the trap closes, and the custom designed LabVIEW software moves the stage to the next sample position. This process then repeats. The laser only fires after reaching a sample position and receiving a trigger pulse from the MS. Multiplexing was used, which we have added into the methods section (line 388). We have added system parameters (line 331-338) in accordance with 10.1002/mas.21585.
A minor comment: the authors may consider reporting the instrument parameters as suggested by Gabelica et al. (10.1002/mas.21585).
We have added instrument parameters in the methods section (line 331-338).
3- Can the authors perform a more in-depth analysis of the data? In the comparison between single-cell and bulk cell analyses, the authors stated that a lot more molecules were detected in single cell experiments. Is there a way to represent them qualitatively? I.e., the average number of molecules per single-cell mass spectrum vs. the number of molecules detected in bulk analysis. A Venn diagram would be nice, but the authors can choose to do it differently. Same comment can be made on last section on variations in the number of saccharides. Can the authors make a 2D plot of intensity vs. cell size?
We thank the author and agree with their first comment. We have added a Venn diagram showing spectral similarities (Figure 5c), and identifications present in both the individual and combined datasets (Figure 5d). Related to the second point, linking signal variation across the cell set to cell size may be problematic as we observed that the entire cell is not ablated (evidenced in figure 2). This means that the total metabolite content measured per cell is not a function of total cell size. Additionally, this is problematic as ion yield is dependent upon the total volume of water within the sampling region, therefore larger cell sizes do not necessary correlate to more signal as the total number of molecules in the sampling area may be less than in a smaller cell. There are a number of variables worth investigating, which we will explore in following studies.
4- Could some of the putative metabolites, or the isomers in Figure 4 be confirmed by LC-MS/MS of bulk homogenates/extracts?
We refer the Reviewer to the assignments in Table 1, which were based on comparing with reference libraries and previous publications. As a next step, we could confirm the identities of these molecules with LC-MS/MS of bulk homogenates/extracts. However, this was proof-of-concept manuscript illustrating the potential power of this system, so it is beyond the scope of this work.
5- There is a problem with the texts in the first paragraph of section 2.3. A few sentences/phrases keep repeating.
We are unclear which section the Reviewer is referring to. Nonetheless, we did reread that section and edited as needed.
Reviewer 3 Report
The authors presented a LAESI-IMS system integrated with an optical microscope for ambient sampling of in situ single cells. The manuscript is generally well written and clearly presented, and the analysis itself is nearly flawless however there are some minor comments:
(1) What types of samples can this system process? (e.g. Living cells, fresh frozen tissues or cells, fixed tissues or cells)
(2) What is the throughput of this system?
Author Response
The authors presented a LAESI-IMS system integrated with an optical microscope for ambient sampling of in situ single cells. The manuscript is generally well written and clearly presented, and the analysis itself is nearly flawless however there are some minor comments:
(1) What types of samples can this system process? (e.g. Living cells, fresh frozen tissues or cells, fixed tissues or cells)
The LAESI-IMS system can process any sample containing water, were it uses water as the ablation matrix. Therefore, LAESI sources can process living and fresh frozen tissue and cells, and some fixed tissue samples, as long as there is sufficient water in the samples. The introductory section (line 49-66) gives a good overview of the technique and its application for cellular analysis.
(2) What is the throughput of this system?
It is difficult to state the overall throughput of our system currently as the cell selection step is currently manual. All aspects of the system have the potential to be automated, therefore can be “high-throughput”. We have added a statement in the conclusion (line 291-293) that reflects this.
Round 2
Reviewer 1 Report
Looks good! Great job addressing the comments.